# Cancer in disguise: a parasite within

Marek Wagner [iD][1][✉] & Shigeo Koyasu [iD][2,3][✉]

## Abstract

Cancer does not simply develop unchecked—it strategically exploits its host with parasitic precision. From immune evasion to tissue remodeling, cancer cells mirror the survival strategies of parasitic helminths. This resemblance suggests that malignant cells have co-opted deeply conserved, evolutionarily honed tactics used by parasites to persist within their hosts. By mimicking helminths, cancer cells may also engage type-2 immune responses, traditionally associated with anti-parasitic defense, as part of the host's attempt to control their expansion. Such parallels could also help explain why type-2 immunity, once considered tumor-promoting, has recently emerged as a potential source of tumoricidal activity. In this Perspective, we explore mechanistic parallels between cancer and helminth infection. Recognizing the parasitic nature of cancer cells not only challenges established models of oncogenesis but also reveals mechanisms that could be leveraged for therapy.

**Keywords** Cancer; Type-2 Immunity; Tumor Microenvironment; Parasites; Anthelmintics

## Introduction

For cancer cells, dissemination from a primary tumor to distant organs marks the transition to malignancy. For many parasitic helminths (or worms), tissue migration is a normal part of their life cycle. In both cases, the journey is perilous, exposing vulnerabilities that can be targeted by the immune system. Type-2 immunity plays a central role in host defense against helminths, which infect nearly a quarter of the global population and impose a major public health burden. Helminths are invertebrates broadly classified as flatworms (platyhelminths, from Greek *platy*, meaning "flat"), which include flukes and tapeworms, and roundworms (nematodes, from Greek *nemato*, meaning "thread").

Much of our understanding of protective immune mechanisms against gastrointestinal helminths comes from studies using rodent models that recapitulate key features of human parasitic infections. One widely used model involves infection with the nematode *Nippostrongylus brasiliensis*. Following subcutaneous injection, third-stage larvae migrate through the bloodstream to the lungs, are coughed up and swallowed, and eventually reach the small intestine, where they feed on epithelial tissue at the base of the villi. This triggers villus remodeling and elicits a strong type-2 immune response. This model was instrumental in identifying group-2 innate lymphoid cells (ILC2s) as an early and essential source of type-2 cytokines, including interleukin (IL)-5 and IL-13 (Moro et al, 2010; Neill et al, 2010). Type-2 cytokines drive the classical "weep and sweep" response by promoting epithelial mucus production and smooth muscle contraction, facilitating worm expulsion. ILC2 function is further supported by IL-2 derived from MHC class-II-dependent interactions with primed CD4 T cells, which subsequently differentiate into T-helper-2 ($T_H2$) cells. These $T_H2$ cells amplify the anti-helminth response by producing the same effector cytokines and contributing to long-term humoral immunity (Zaiss et al, 2024).

Beyond their anti-parasitic role, type-2 immunity has more recently been linked to tumor control (Wagner et al, 2025). One suggested mechanism involves the induction of eosinophilia by ILC2s and $T_H2$ cells (Wagner et al, 2020; Wagner et al, 2025). During helminth infection, eosinophilia is most pronounced in the early stages - during larval migration - when tissue damage is greatest. As the parasites mature and the infection becomes apparent, the host immune responses are suppressed through regulatory pathways that enable parasite persistence. Cancer cells appear to exploit similar immunoregulatory mechanisms to evade immune surveillance and sustain their growth. Consequently, helminths and cancer cells persist for months or years, often without acute symptoms.

In this Perspective, we explore mechanistic parallels between helminth infection and cancer, highlighting shared features of tissue migration, immune modulation, and evasion. Understanding these convergent pathways may reveal novel targets for therapies.

## Parallels between helminth infections and cancer biology

Studies of helminth and cancer cell invasion mechanisms revealed shared strategies (Fig. 1). Helminths penetrate host tissue and disseminate by secreting multiple classes of proteases. For example, the infective third-stage larvae of *Necator americanus*—a soil-transmitted hookworm nematode responsible for ~85% of all hookworm infections—invade human skin by deploying aspartic, serine, and metalloproteinases, which degrade critical components of the extracellular matrices (ECMs), including collagen, elastin, and fibronectin (Brown et al, 1999). Similarly, cancer cells such as melanoma—which originates from melanocytes in the basal layer of the epidermis and has a high propensity to metastasize—produce a broad range of proteolytic enzymes, particularly matrix metalloproteinases (MMPs). Importantly, MMP7, MMP11, and MMP14 have been associated with poor overall survival in melanoma patients (Wu et al, 2025). These enzymes concentrate in invadopodia, actin-

[1]Innate Immunity Research Group, Life Sciences and Biotechnology Center, Łukasiewicz Research Network – PORT Polish Center for Technology Development, Wrocław, Poland. [2]RIKEN Center for Integrative Medical Sciences (IMS), Yokohama, Japan. [3]National Institutes for Quantum Science and Technology (QST), Chiba, Japan.
[✉]E-mail: marek.wagner@port.lukasiewicz.gov.pl; shigeo.koyasu@riken.jp
https://doi.org/10.1038/s44318-025-00691-y | Published online: 13 January 2026

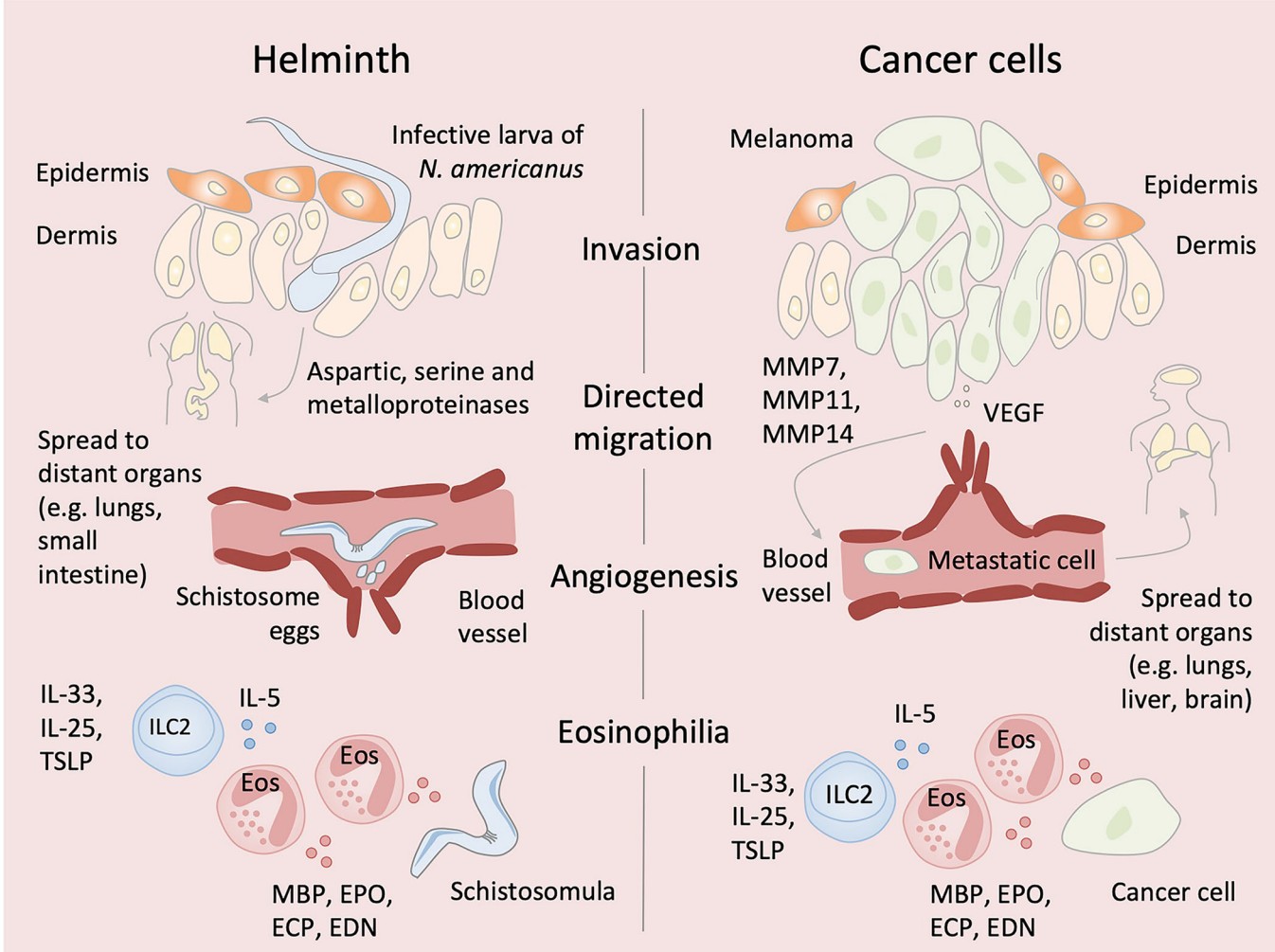

**Figure 1. Shared biological strategies of helminths and cancer cells.**

Helminths such as *Necator americanus* larvae secrete aspartic, serine, and metalloproteinases to degrade extracellular matrix (ECM) components, whereas cancer cells such as melanoma cells use invadopodia enriched with MMP7, MMP11, and MMP14 to breach ECM barriers. Both exhibit reduced cytoskeletal stiffness, facilitating tissue penetration. Helminths follow defined migratory routes to colonize permissive niches; melanoma cells disseminate to preferred metastatic sites. Schistosome eggs and tumor cells both induce angiogenesis to sustain growth. Helminths and tumors engage type-2 immunity; IL-33, IL-25, and TSLP activate ILC2s, which secrete IL-5 to recruit and activate eosinophils. Eosinophils release cytotoxic proteins—MBP, ECP, EDN, EPO—that damage helminth tegument and can kill tumor cells. ECM extracellular matrix, MMP matrix metalloproteinase, VEGF vascular endothelial growth factor, TSLP thymic stromal lymphopoietin, ILC2 group-2 innate lymphoid cell, MBP major basic protein, ECP eosinophil cationic protein, EDN eosinophil-derived neurotoxin, EPO eosinophil peroxidase.

rich protrusions that facilitate tissue penetration, much like parasite invasion (Legrand et al, 2023).

Atomic force microscopy has revealed that both cancer cells—metastatic ones—and helminths, especially at larval stages, generally exhibit low mechanical stiffness (Cross et al, 2007; Dantas et al, 2024). Cancer cells, due to cytoskeletal and nuclear adaptations, can be over 70% softer than their benign counterparts (Cross et al, 2007). This shared mechanical softness facilitates tissue invasion and migration, suggesting that reduced cellular stiffness may represent a convergent biomechanical strategy used by both metastatic cancer cells and invasive helminth larvae to overcome physical barriers within host tissues.

Once inside the host, helminths exhibit highly specialized migratory behaviors—similar to cancer cells—tailored to colonize permissive niches. For example, *N. americanus* larvae penetrate skin capillaries and enter the bloodstream, travel to the lungs, and eventually reach the small intestine, where they latch onto the intestinal mucosa using specialized cutting plates and feed on blood (Loukas et al, 2016). Similarly, cancer cell migration is not random but reflects a selective tropism for specific tissues.

Specifically, melanoma cells invade the surrounding dermis, enter nearby lymphatic vessels, and migrate to sentinel lymph nodes. From there, they disseminate through the bloodstream and preferentially metastasize to organs such as the lungs, liver, bones, and brain (Tasdogan et al, 2025). This directional migration is guided by both soluble factors (e.g., growth factors) and insoluble components (e.g., ECMs).

Helminths also promote angiogenesis, a process similarly exploited by cancer cells to support tumor growth and progression. Schistosomes, parasitic flatworms (or blood flukes) that reside within blood vessels,

produce eggs that appear to stimulate angiogenesis to support their life cycle — potentially by enhancing blood flow in response to vessel occlusion caused by the worms or their eggs. Intact eggs of *Schistosoma mansoni* and their soluble egg antigen (SEA) have been shown to induce proliferation of human umbilical vein endothelial cells (HUVECs) (Freedman and Ottesen, 1988). SEA has also been reported to upregulate VEGF expression in HUVECs (Loeffler et al, 2002). Furthermore, patients with schistosomiasis have been found to exhibit significantly elevated serum VEGF levels compared to healthy individuals (Shariati et al, 2011). Moreover, increased vascularization has been observed in cervicovaginal mucosal biopsies containing *Schistosoma haematobium* eggs, compared to egg-free control tissues (Jourdan et al, 2011).

While cancer cells critically depend on the vascular system to access oxygen and nutrients such as glucose, certain helminths —including hookworms and schistosomes —actively consume host blood as a direct source of metabolic substrates. For example, *N. americanus* secretes anticoagulants at its attachment site in the intestinal mucosa and within its digestive tract to feed on blood, causing iron-deficiency anemia (IDA), particularly in vulnerable individuals like young children, pregnant women, or those with heavy infections (Loukas et al, 2016). However, most of the host's blood loss arises not from the worm's direct consumption, but from persistent bleeding at the sites of mucosal attachment. Although adult hookworm consumes only about 1 μl of blood per day, infections with *N. americanus* can result in cumulative blood losses exceeding 1 ml per day (Loukas et al, 2016). Similarly, IDA is often observed in cancer patients, particularly those with gastrointestinal malignancies, where chronic, low-grade bleeding from the tumor bed contributes to the development of anemia (Almilaji et al, 2021).

Another prominent feature of helminth infection is eosinophilia, which has been documented in controlled human infection studies with *N. americanus*. Elevated eosinophil counts were observed as early as four weeks post-infection—during the larval migration phase, when tissue damage is most extensive—preceding the arrival of the worms in the intestine (Loukas et al, 2016). The release of epithelial-derived alarmins, such as IL-25, IL-33, and thymic stromal

lymphopoietin (TSLP), activates ILC2s, which together with $T_H2$ cells produce effector cytokines including IL-5 known to promote eosinophil activation. Furthermore, $T_H2$ cells support long-term humoral immunity by inducing B cells to secrete IgE antibodies through the production of IL-4 and IL-13. Notably, B cells have also been shown to produce IL-2, which promotes ILC2 expansion and sustains an IL-5-dependent, eosinophilic axis (Whyte et al, 2022). The ability of eosinophils to kill helminths has been demonstrated in vitro using *S. mansoni* schistosomula in the presence of specific antibodies (Capron et al, 1979). Upon activation, eosinophils release cytotoxic granule proteins, including major basic protein (MBP), which exerts potent membrane-disruptive effects due to its high cationic charge. Other granule proteins with cytotoxic activity include eosinophil cationic protein (ECP), eosinophil-derived neurotoxin (EDN), and eosinophil peroxidase (EPO) (Wagner et al, 2025). Granule proteins such as MBP have been observed to progressively damage the tegument (outer layer) of antibody-coated *S. mansoni* schistosomula, resulting in its detachment and exposure of the underlying musculature, which is subsequently degraded and phagocytosed (Butterworth et al, 1979). In the *Heligmosomoides polygyrus* infection model, *ΔdblGATA* mice— which lack eosinophils—exhibit a higher worm burden compared to wild-type controls, suggesting a role for eosinophils in controlling primary infection (Hewitson et al, 2015). Nevertheless, in vivo evidence indicates that eosinophils mainly enhance the immune response rather than serve as key effectors in clearing adult intestinal worms.

Interestingly, lysates from eosinophils expressing MBP have been shown to exert cytotoxic effects on B16-OVA murine melanoma cells (Mattes et al, 2003). Similarly, co-culture of human eosinophils with Colo-205 colorectal carcinoma cells led to the release of ECP, EDN, TNF-α, and granzyme A, all contributing to cancer cell death (Legrand et al, 2010). These findings are supported by in vivo studies. Depletion of eosinophils using anti-SIGLEC-F antibodies enhanced melanoma growth and metastasis (Lucarini et al, 2017). In a model of methylcholanthrene (MCA)-induced fibrosarcoma, IL-5 transgenic mice with elevated eosinophil levels demonstrated reduced tumor incidence. Conversely, mice

deficient in the eosinophil chemoattractant CCL11 (eotaxin-1), or lacking eosinophils altogether ($Il5^{-/-}Ccl11^{-/-}$ and $ΔdblGATA$ mice) exhibited increased tumor susceptibility, highlighting the protective role of eosinophils (Simson et al, 2007). Finally, eosinophilia has also been proposed as a potential prognostic marker for patients with various tumors, including oral squamous cell carcinoma, colorectal carcinoma and melanoma (Wagner et al, 2025).

Given that certain helminths—such as *S. haematobium*, *Clonorchis sinensis*, and *Opisthorchis viverrini*—are classified as group 1 biological carcinogens for promoting cancers like squamous cell carcinoma of the urinary bladder and cholangiocarcinoma (Wagner et al, 2025), eosinophils may have partly evolved to lower the cancer risk associated with chronic parasitic infections.

## Mechanisms to evade immune surveillance

Studies of immune responses to helminths and cancer cells have also revealed shared immunomodulatory strategies for evading immune surveillance (Fig. 2). Helminths epitomize evolutionary finesse in modulating host immunity. A key strategy of these parasites is the recruitment and expansion of CD4$^+$ regulatory T cells (Tregs), which suppress anti-parasitic immunity by inhibiting the activation, proliferation, and effector functions of conventional T cells through IL-10, TGF-β, and contact-dependent mechanisms (McManus and Maizels, 2023).

In murine models of infection with *H. polygyrus*, *Litomosoides sigmodontis*, *Schistosoma japonicum*, and *Strongyloides ratti*, Tregs play a critical role in facilitating parasite persistence by dampening immune responses aimed at eliminating the infecting organisms (McManus and Maizels, 2023). This pattern also extends to humans, especially those who remain asymptomatic or hyporesponsive during infection, where a clear association between helminth burden and Treg activity has been observed. For example, *S. mansoni* induces robust Treg expansion - a response that is reversed following treatment with praziquantel, an anti-schistosomal drug (Schmiedel et al, 2015). Helminths manipulate the host immune system by secreting immunomodulatory proteins. For example, *H. polygyrus* produces Hp-TGM, a TGF-β mimic that drives Treg expansion in both mice and

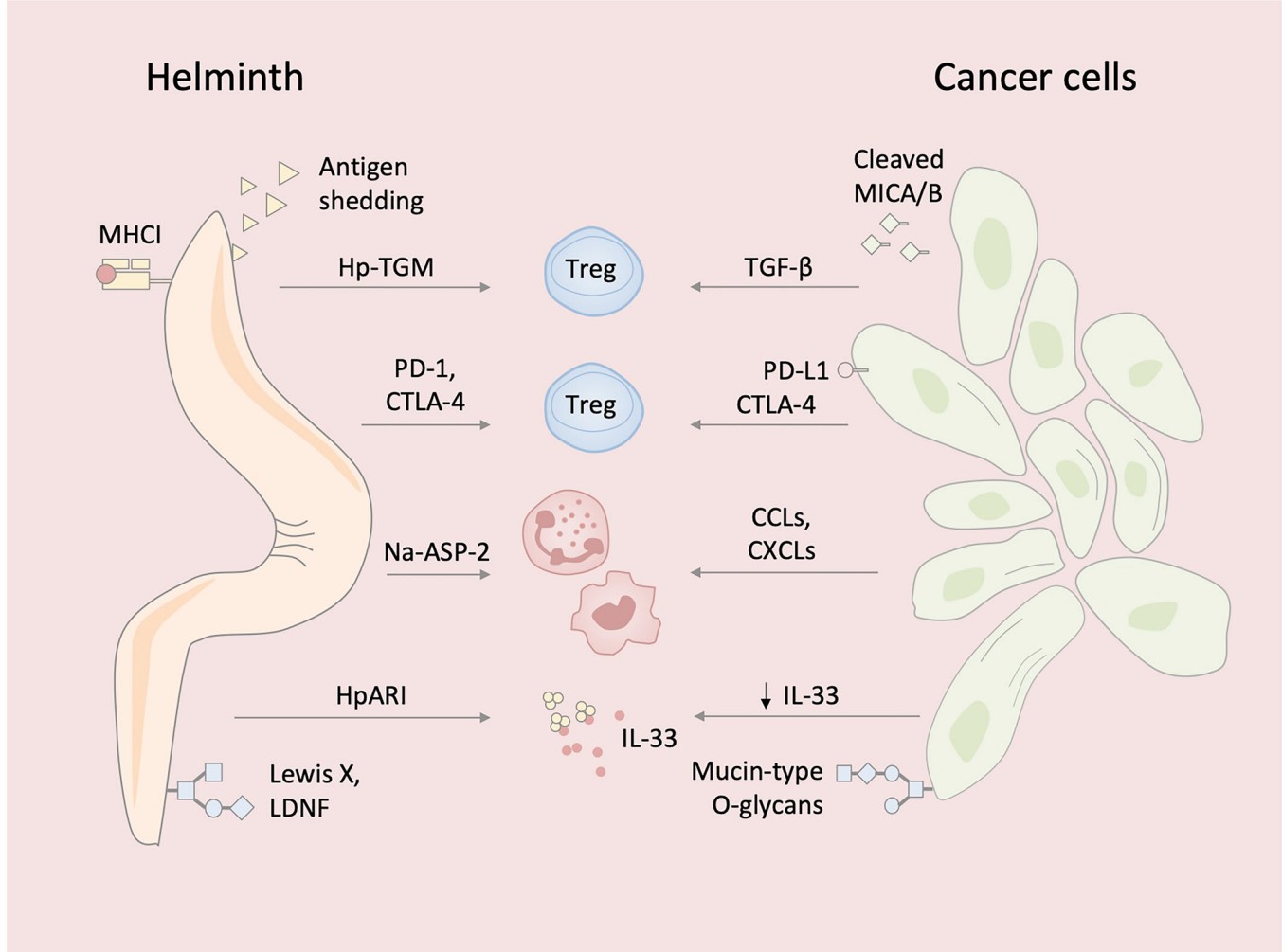

**Figure 2.  Shared immune evasion strategies between helminths and cancer cells.**

Helminths evade host immunity through multiple mechanisms, including expansion of regulatory T cells (Tregs) via immunomodulatory cytokines and TGF-β mimics (e.g., Hp-TGM), upregulation of immune checkpoint receptors (PD-1, CTLA-4), shedding of antigenic surface molecules, and molecular mimicry through host-like glycans (e.g., Lewis X, LDNF) or acquisition of host MHC class-I glycoproteins. They also modulate innate immunity by mimicking chemokines (e.g., Na-ASP-2) to recruit myeloid cells and by secreting inhibitors (e.g., HpARI) that block the release of epithelial alarmins such as IL-33. Tumors employ analogous strategies, including recruitment of Tregs, expression of immune checkpoints, antigen shedding - such as proteolytic release of MICA and MICB leading to NKG2D downregulation - MHC class-I downregulation, infiltration of myeloid-derived cells, and suppression of IL-33 signaling. These parallels highlight convergent evolution in immune evasion mechanisms between helminths and cancer cells. Treg regulatory T cell, TGF-β transforming growth factor beta, Hp-TGM *Heligmosomoides polygyrus* TGF-β mimic, PD-1 programmed cell death protein 1, CTLA-4 cytotoxic T-lymphocyte-associated protein 4, LDNF Lacto-N-fucopentaose III, MHC class-I major histocompatibility complex class I, Na-ASP-2 *Necator americanus* Ancylostoma Secreted Protein-2, HpARI *Heligmosomoides polygyrus* alarmin release inhibitor, IL-33 interleukin-33, MICA/MICB MHC class-I chain-related proteins A and B.

humans (Johnston et al, 2017). Cancer cells adopt a strikingly similar strategy. In several human malignancies—including melanoma, cervical, renal, lung, hepatocellular, and gastric cancers—increased frequencies of tumor-infiltrating Tregs correlate with poor prognosis and reduced overall survival (Shang et al, 2015).

To reinforce local immunosuppression, helminths upregulate immune checkpoint molecules, particularly programmed cell death protein 1 (PD-1) and cytotoxic T-lymphocyte-associated antigen 4 (CTLA-4) on immune cells, including Tregs. In the case of hepatic alveolar echinococcosis caused by *Echinococcus multilocularis*, elevated levels of PD-1 and CTLA-4 have been observed on Tregs isolated from the liver tissue of infected individuals (Sun et al, 2024). In corresponding murine models of *E. multilocularis* infection, increased expression of these checkpoint molecules has been found on splenic and peritoneal CD4⁺ T cells (Sun et al, 2024). In cancer, PD-1 expression has been shown to correlate with T-cell exhaustion in various tumors (Simon and Labarriere, 2017), and CTLA-4 is often overexpressed on tumor-infiltrating Tregs (Ding et al, 2024), further promoting local immunosuppression within the tumor microenvironment.

From a different perspective, the infective third-stage larvae of *Dirofilaria immitis* evade immune recognition by actively shedding a substantial portion (10–20% per day) of their surface peptides, thereby reducing their antigenic visibility to the host

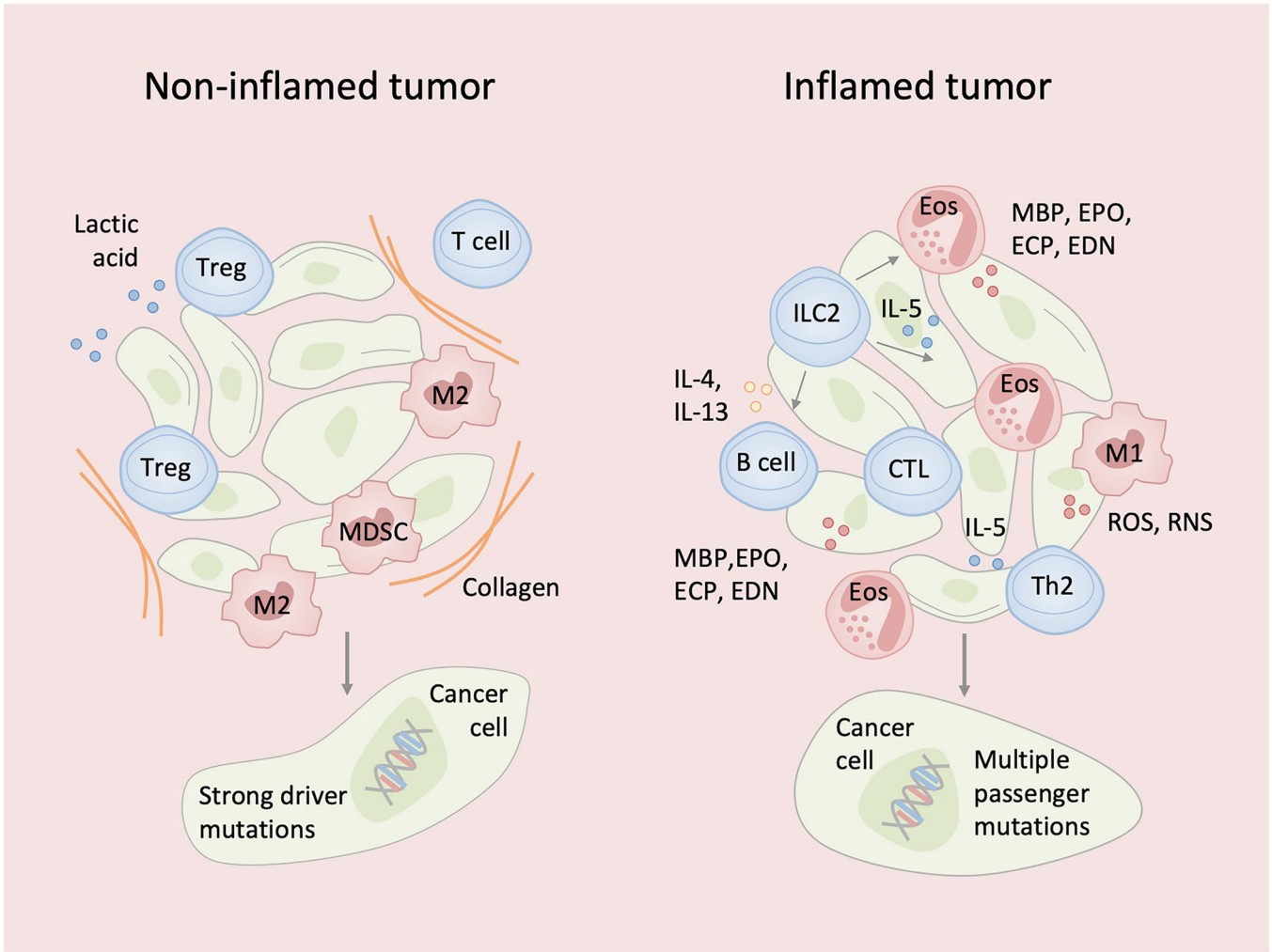

**Figure 3.   Impact of driver and passenger mutations on tumor immune phenotype.**

(Left panel) Tumors with strong oncogenic driver alterations (e.g., *RHOA* mutations in gastric cancer, *EGFR* mutations in non-small-cell lung cancer) develop an immunosuppressive, non-inflamed tumor microenvironment (TME). These mutations directly suppress antitumor immunity by inhibiting cytotoxic T cells (CTLs), recruiting regulatory T cells (Tregs) and M2-like macrophages, increasing extracellular matrix (ECM) deposition, and producing lactate that promotes immunosuppressive macrophage polarization while inhibiting effector cell functions. In this context, type-2 immune responses may promote tumor growth through wound-healing-like mechanisms and enhanced tissue remodeling. (Right panel) Passenger mutation–rich tumors accumulate numerous genetic alterations, leading to increased tumor neoantigen presentation. This immunogenicity promotes an inflamed TME with infiltration of CTLs and ILC2s. Here, IL-25, IL-33, and TSLP activate ILC2s to produce IL-5, which recruits eosinophils. Eosinophils release cytotoxic proteins (e.g., MBP, ECP, EDN, EPO) that can directly kill tumor cells and facilitate further T-cell infiltration, enhancing tumor control. CTL cytotoxic T cell, Treg regulatory T cell, Th2 T-helper-2 cell, TSLP thymic stromal lymphopoietin, ILC2 group-2 innate lymphoid cell, EOS eosinophil, MBP major basic protein, ECP eosinophil cationic protein, EDN eosinophil-derived neurotoxin, EPO eosinophil peroxidase.

immune system (Ibrahim et al, 1989). Cancer cells, such as multiple myeloma cells, escape immune surveillance by proteolytically shedding MHC class-I chain-related proteins A and B (MICA/B)—stress-induced ligands recognized by natural killer (NK) cells. This shedding leads in downregulation of the activating receptor NKG2D on NK cells, impairing their cytotoxic activity and enabling tumor immune evasion (Tahri et al, 2024).

Molecular mimicry further contributes to immune evasion. Many helminths express glycoconjugates that closely resemble host antigens—such as Lewis X and LDNF— thereby promoting immune tolerance (Naus et al, 2003). Tumors similarly exploit aberrant glycosylation through the expression of mucin-type O-glycans associated with fetal development, a phenomenon known as oncofetal glycosylation (Freire-de-Lima et al, 2011). Additionally, *S. mansoni* has been shown to incorporate intact host MHC class-I glycoproteins into its surface to mimic self, whereas cancer cells evade cytotoxic T-cell recognition by downregulating or altering their own MHC class-I expression (Simpson et al, 1983).

For both helminths and cancer cells, carbohydrate serves as an essential energy source. Adult parasitic helminths such as schistosomes rely primarily on anaerobic metabolism, even in the presence of oxygen (Bexkens et al, 2024). Similarly, cancer cells preferentially undergo anaerobic glycolysis—converting glucose to lactate under oxygen-rich conditions (the Warburg effect). Tumor-derived lactate strongly suppresses immune effectors, including CD8[+]

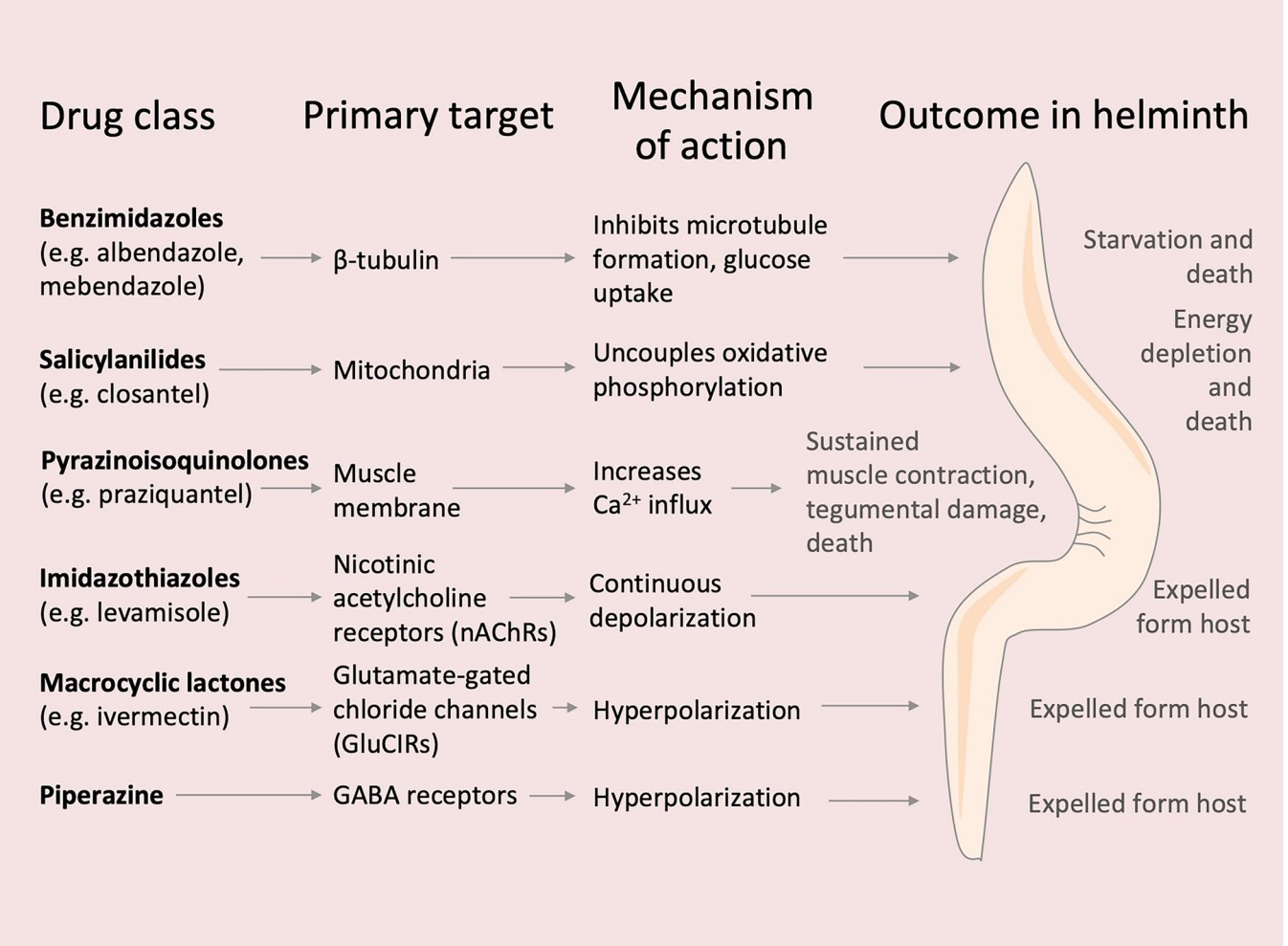

**Figure 4. Anthelmintic drug classes, molecular targets, and functional outcomes.**

Overview of six major anthelmintic drug classes, showing representative compounds, primary molecular targets, modes of action, and final effect on worms (i.e., death or expulsion). Notably, several of these parasite-directed targets (e.g., β-tubulin, mitochondrial metabolism, ion channels) are also being explored as therapeutic vulnerabilities in cancer, highlighting a growing overlap between anti-parasitic and antitumor pharmacology.

T cells and innate lymphoid cells (ILCs), by impairing their function and viability (Marciniak and Wagner, 2023; Wagner et al, 2020). While helminths engage metabolic pathways, such as amino acid and lipid metabolism, to generate immunoregulatory molecules, it remains unclear whether they secrete lactate extracellularly to modulate host immunity.

To facilitate tissue invasion, infective third-stage larvae of *N. americanus* secrete Ancylostoma Secreted Protein-2 (Na-ASP-2), which induces a neutrophil- and monocyte-rich infiltrate, likely through CC-chemokine mimicry. The resulting inflammation and edema increase tissue permeability, while neutrophil dominance protects the parasite from cytotoxic effectors such as eosinophils or NK cells (Bower et al, 2008). Similarly, cancer cells recruit myeloid cells, including neutrophils and macrophages. Tumor-derived colony-stimulating factor 1 (CSF-1) and CCL2 promote the recruitment and polarization of macrophages toward an M2 phenotype, which supports tumor growth and progression by enhancing angiogenesis, facilitating tumor cell extravasation, and supporting metastasis through factors such as VEGFA. Neutrophils can likewise acquire an alternatively activated, pro-tumor N2 phenotype that contributes to metastasis, angiogenesis, and immunosuppression. Tumor-derived CSF-2 has been shown to drive this polarization, and PD-L1-expressing neutrophils can suppress T-cell activity (El-Naccache et al, 2021, for a comprehensive review).

Epithelial barriers, including the skin, serve as critical sentinels of immune activation. In response to parasitic damage, epithelial cells release IL-33, an alarmin cytokine that potently activates ILC2s to mount rapid type-2 immune responses (Moro et al, 2010). Exogenous administration of IL-33 enhances anti-parasitic immunity, underscoring its central role. To subvert this pathway, *H. polygyrus* secretes alarmin release inhibitor (HpARI), a protein that binds both IL-33 and nuclear DNA, effectively trapping IL-33 within necrotic cells and preventing its extracellular release and subsequent initiation of type-2 immune responses (Osbourn et al, 2017). In parallel,

*H. polygyrus* deploys small RNA-containing extracellular vesicles to suppress the transcription of IL-33 receptor (Buck et al, 2014). Notably, clinical samples from patients with prostate and renal carcinomas showed reduced IL-33 expression during the transition from primary to metastatic disease, suggesting convergent strategies of immune evasion between parasites and tumors (Saranchova et al, 2016). Interestingly, a positive correlation between the expression of *IL33* and prolonged overall survival has been identified in melanoma patients (Wagner et al, 2020). However, this association does not appear to extend to all tumor types. In gastric cancer, for example, *IL33* expression does not correlate with overall survival (Hu et al, 2017). Moreover, in another study IL-33/mast cell activation gene signature has been associated with poorer overall survival in patients with intestinal-type gastric cancer (Eissmann et al, 2019). These findings suggest that the antitumor effects of type-2 immunity may be tumor-type specific.

Human cancers are broadly classified into inflamed and non-inflamed types (Fig. 3). According to "the immunogenomic cancer evolution" hypothesis (Kumagai et al, 2024), tumors driven by strong oncogenic mutations (e.g., *RHOA* mutations in gastric cancer) tend to develop a non-inflamed, immunosuppressive microenvironment. In contrast, tumors lacking such strong drivers may accumulate immunogenic passenger mutations and often present with inflamed microenvironments. In this context, type-2 immune responses may contribute to tumor suppression.

## Repurposing anthelmintic drugs for cancer therapy

Although helminth infections impose a major burden on public health, currently approved anthelmintic drugs fall into only a few major chemical classes, each acting on highly conserved molecular targets (Fig. 4). Benzimidazoles (e.g., albendazole, mebendazole) bind selectively to parasite β-tubulin, preventing microtubule polymerization and thereby inhibiting glucose uptake, ultimately leading to energy depletion and death. Salicylanilides, such as closantel, disrupt mitochondrial function by uncoupling oxidative phosphorylation, collapsing the proton gradient required for ATP synthesis, and inducing metabolic failure. Pyrazinoisoquinolones (e.g., praziquantel)

trigger a rapid calcium influx into parasite muscle cells, leading to sustained muscle contraction and tegumental damage that exposes the parasite to host immune attack. Imidazothiazoles (e.g., levamisole) act as agonists at nicotinic acetylcholine receptors (nAChRs) on nematode muscle, causing sustained depolarization and rigid paralysis. Macrocyclic lactones, such as ivermectin, bind glutamate-gated chloride channels (GluClRs), inducing hyperpolarization, flaccid paralysis, and eventual expulsion by host peristalsis. Piperazine, in contrast, functions as a GABA receptor agonist at inhibitory neuromuscular synapses, also leading to flaccid paralysis through suppressed excitatory signaling, which results in worm expulsion (Nixon et al, 2020).

The recognition of functional and metabolic parallels between cancer cells and parasitic helminths opens up new avenues for repurposing anthelmintic drugs. Among these, mebendazole has shown compelling preclinical activity across multiple malignancies, including pancreatic, lung, thyroid, breast, colorectal, skin, brain, and meningeal cancers (Joe et al, 2022). Originally developed to disrupt microtubule formation in parasitic worms by targeting β-tubulin, mebendazole exerts similar effects in tumor cells, interfering with cell division—an approach shared by conventional chemotherapeutics like paclitaxel and vincristine. Beyond its cytostatic effects, mebendazole has been reported to impair cancer cell invasiveness and metastatic potential. For example, it reduces integrin β4 expression and suppresses cancer stem-like traits in breast cancer models, correlating with reduced tumor growth and metastatic dissemination (Joe et al, 2022). Additional antitumor mechanisms of mebendazole include inhibition of angiogenesis via VEGFR-2 blockade (validated using HUVEC-based angiogenesis assays), induction of apoptosis through BCL-2 modulation in chemoresistant melanoma, and activation of caspase-3-dependent cell death pathways in lung cancer cell lines (Pantziarka et al, 2014). Mebendazole also induces G2/M cell-cycle arrest followed by apoptosis in multiple lung cancer models (Pantziarka et al, 2014). Isolated clinical case reports further describe durable responses in patients with metastatic adrenocortical and colorectal cancer treated with mebendazole (Pantziarka et al, 2014).

Collectively, these findings highlight the importance of further investigating the

repositioning of anthelmintic drugs, including mebendazole, in oncology to validate their therapeutic potential and to define safe protocols for clinical use.

## Conclusion

Consistent with the traditional notion that type-2 immunity protects from helminth infections, emerging evidence suggests that type-2 immune responses also constrain tumor growth and progression (Wagner et al, 2025). Cancer cells appear to mirror the survival strategies of parasitic helminths, suggesting that malignant cells have co-opted evolutionarily conserved mechanisms that parasites use to persist within their hosts. By mimicking parasitic behavior, cancer cells may also engage type-2 immunity as a host response. However, prolonged activation of type-2 immunity—as seen in chronic helminth infections—can induce counterregulatory mechanisms, including Treg expansion, limiting immune-mediated tissue damage but potentially facilitating immune escape. This immune dampening, likely co-opted by both parasites and tumors, represents a major challenge for sustained antitumor immunity. Given the complexity of helminth secretomes, their intricate host interactions, and the sophistication of the immune responses they elicit, many helminth-derived immunomodulators likely remain undiscovered. A deeper understanding of the evolutionary parallels between helminths and cancer cells could uncover new immunotherapeutic targets and translate advances in parasitology into innovative cancer treatments.

## Peer review information

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

## Acknowledgements

This work was supported by the Narodowe Centrum Nauki (NCN) SONATA BIS12 project no. 2022/46/E/NZ6/00131 to MW. The authors thank H Nishikawa for insightful discussion on the immuno-genomic cancer evolution hypothesis.

## Author contributions

**Marek Wagner**: Conceptualization; Writing—original draft. **Shigeo Koyasu**: Conceptualization; Writing—review and editing.

## Disclosure and competing interests statement

The authors declare no competing interests.

