## [Peer Review File · The EMBO Journal]

Cancer in disguise: a parasite within

Marek Wagner and Shigeo Koyasu

Corresponding author(s): Shigeo Koyasu (shigeo.koyasu@riken.jp) , Marek Wagner (marek.wagner@port.lukasiewicz.gov.pl)

Review Timeline:

Submission Date:	17th Sep 25
Editorial Decision:	25th Oct 25
Revision Received:	8th Nov 25
Accepted:	24th Nov 25

Editor: Ioannis Papaioannou

Transaction Report:

Dear Dr. Koyasu,

Thank you for submitting your perspective (EMBOJ-2025-122490-T) to The EMBO Journal for our consideration, and for your patience during peer review. Your manuscript has been seen by three experts in the field, and we have now received their comments, which are appended below.

I am glad to say that, as you will see, all three referees are supportive of your manuscript, mentioning that it will likely be of interest not only to experts in the respective fields but also, beyond them, to our broad readership. They have only a few suggestions for relatively minor edits to strengthen your manuscript further, which I would like to invite you to address in a revised version of your manuscript, as well as in a point-by-point response to the referees' comments and suggestions.

The best-fitting article type for your manuscript at our journal is that of a Perspective. A Perspective is a scholarly review article format dedicated to scientific topics within the scope of the journal. Perspectives should be focused on specific areas of research with notable recent or prospective progress; new approaches or techniques can form a focus. A Perspective article should "set the scene" based on recent developments with an emphasis on future directions of the field of study. Perspectives can take a personal point of view, but should emphasize reported facts and testable hypotheses over speculation and opinion. Perspectives contain an abstract and introduction, and the main text should have subheadings. The format is flexible, but we aim for succinct and focused discussions to engage a broad readership. We recommend no more than 4 figures, 5000 words and 50 references. We encourage the use of Text boxes to structure the article and summarize key milestones related to the topic of discussion. This provides a foundation for the more forward-looking main body of the text. In addition to the instructions above, we also kindly request that you address the following minor specific points in your revised manuscript:

- The co-corresponding authors must be indicated (e.g. using asterisks or similar) in the authors' list on the title page.
- Please note that it is mandatory to include a conflict-of-interest statement, with heading "Disclosure and competing interests statement". Employment in a biotechnology company should be stated in this section. You can find more information in our author guidelines: <https://www.embopress.org/page/journal/14602075/authorguide#conflictsofinterest>.
- Please note that up to 5 keywords (preferably broad terms to enhance online search engine discoverability of your Perspective) can be listed (you currently list 6 keywords); the keyword list should be provided below the Abstract of the revised manuscript.
- We noticed that you have listed 69 references in your list, but a Perspective cannot have more than 50. We thus kindly request that you reduce the number of References to 50 removing non-essential citations.
- Please also note that the format of the References is not correct; they should appear in alphabetical order, and the names of the first 10 co-authors of each, followed by "et al." should be provided for each reference. Please remove the existing additional comments/highlights from some of your references. For more information on our References format, please see our guide to authors: <https://www.embopress.org/page/journal/14602075/authorguide#referencesformat>.
- The order of the manuscript sections should be corrected as follows: Title page - Abstract - Keywords - Introduction - Acknowledgements - Disclosure and Competing Interests Statement - References - Figure Legends.

Please also note that as part of the EMBO publications' Transparent Editorial Process, The EMBO Journal publishes online a Peer Review File along with each accepted manuscript. This File will be published in conjunction with your paper and will include the referee reports, your point-by-point response and all pertinent correspondence relating to the manuscript. You can opt out of this by letting the editorial office know (contact@embojournal.org).

We look forward to seeing a final version of your Perspective as soon as possible. Please let us know if you have any questions and use this link to submit your revision: <https://emboj.msubmit.net/cgi-bin/main.plex>.

Best regards,

Ioannis

Referee #1:

In this perspective Drs. Wagner and Koyasu discuss parallels of the cellular and molecular immune cascade elicited upon helminth infection with the response during cancer and how this could be leveraged for novel treatment strategies. The discussion points are elegantly put together, the text flows nicely, the graphs provided highlight several key arguments. Overall, this manuscript provides a refreshing new perspective that will be stimulating not only to the experts in the fields but also to the broad readership of EMBO.

I highly recommend accepting this manuscript for publication. I only suggest some minor edits to strengthen the work.

1. In the introduction it would be beneficial if the authors could add the diverse sources of IL-2 driving ILC2 according to a recent publication (PMID: 35699942).

2. Two aspects that might be beneficial to include (at least briefly) in this nice perspective are the roles of ILC2 in tumorigenesis vs anti-helminth immunity (immunosuppressive ILC2 vs anti-tumor ILC2; e.g. PMID: 36323785, 38211590, 39814891, 39721022), as well as the contribution of CD8+ T cells (PMID: 30361537, 32624246, 36524995, 37612281).

I congratulate the authors on this refreshing perspective.

Referee #2:

In this commentary, Dr. Marek Wagner and Shigeo Koyasu described the similarity of helminth infection and cancer development regarding tissue invasion, tissue remodeling, and immune cell suppression that can result from persistent cancer progression and helminth infection. The authors noted that both helminth and cancer cells secrete multiple classes of proteases, including metalloproteinases, to degrade critical components of extracellular matrices (ECMs) for their tissue evasion and migration; both promoted angiogenesis, and used blood flow as oxygen and nutrients intake that usually caused bleeding and iron-deficiency anemia; both can develop a type 2 immune response microenvironment including elevations in IL-33, TSLP, IL-4/IL-13, and eosinophilia. Eosinophils can play an important role against helminth infection and cancer development through secretion of toxic enzymes such as major basic protein (MBP).

The authors also summarized the mechanisms that helminth and cancer cells used to evade immune surveillance including activation and expansion of Treg cells; upregulation of immune checkpoint molecules, such as PD-1 and CTLA-4; as well as secreted immunomodulatory proteins such as TGF-beta mimic from *H. polygyrus*; and active shedding of surface peptides by helminth and tumors. Finally, the authors summarized the study of anthelmintic drugs for cancer therapy potential based on the recognition of functional and metabolic parallels between cancer cells and parasitic helminths. For example, they discuss the use of mebendazole, a benzimidazole derivatives, for therapies of multiple cancers.

Although this is an excellent start, the authors need to broaden their comparisons of immune cells in helminth infection and cancer. In particular, myeloid cells including macrophages and neutrophils, need to be discussed. Papers, including recent reviews, by Loke, Maizels (review JEM 2023), Allen, Gause, and Pearce need to be included to provide a more in-depth and more balanced discussion. Of note, a paper published in 2021 in Trends In Immunology (10.1016/j.it.2020.11.006) also discussed similarities between the immune response to helminth infection and cancer. This manuscript, among others, needs to be discussed in the context of this current perspective.

As such this commentary, although of considerable interest, needs significant revisions before it is ready for publication.

Referee #3:

This review compares similarities in the immune response that can be elicited by both cancer and parasitic worm infections. This is an interesting comparison as the review notes that both cancer cells and worms take advantage of similar mechanisms to facilitate its survival and evade immune detection. Given the rapid proliferative capacity of cancer cells and their acquisition of mutations, it is understandable how there is evolutionary convergence for cancer cells to take advantage of similar weaknesses in the immune system as worms. I have included a few comments to hopefully help improve the review.

- Page 8 end of first paragraph, is something missing? This sentence looks like it was cut short.
- The text associated with Figure 3 seems to be missing. Maybe this got accidentally deleted?
- A figure showing how anthelmintic drugs can target both helminths and cancer cell pathways may help illustrate how therapeutic targets identified against worms can also have anti-tumour effects.

EMBOJ-2025-122490-T

Point-by-point responses:

We greatly appreciate the Referee's time and constructive feedback. Revising the manuscript according to these comments has significantly improved its clarity and impact.

Referee comments and our response:**Referee #1:**

In this perspective Drs. Wagner and Koyasu discuss parallels of the cellular and molecular immune cascade elicited upon helminth infection with the response during cancer and how this could be leveraged for novel treatment strategies. The discussion points are elegantly put together, the text flows nicely, the graphs provided highlight several key arguments. Overall, this manuscript provides a refreshing new perspective that will be stimulating not only to the experts in the fields but also to the broad readership of EMBO. I highly recommend accepting this manuscript for publication. I only suggest some minor edits to strengthen the work.

Response: Thank you for your thoughtful and positive evaluation of our manuscript. We appreciate your helpful suggestions.

1. In the introduction it would be beneficial if the authors could add the diverse sources of IL-2 driving ILC2 according to a recent publication (PMID: 35699942).

Response: We agree and have now incorporated this information in the paragraph entitled "Parallels between helminth infections and cancer biology."

2. Two aspects that might be beneficial to include (at least briefly) in this nice perspective are the roles of ILC2 in tumorigenesis vs anti-helminth immunity (immunosuppressive ILC2 vs anti-tumor ILC2; e.g. PMID: 36323785, 38211590, 39814891, 39721022), as well as the contribution of CD8+ T cells (PMID: 30361537, 32624246, 36524995, 37612281).

Response: We appreciate this insightful comment. However, due to the journal's strict reference limit of 50 citations, which we have already exceeded in the initial

submission with 69 references, it is not possible to expand the manuscript further without removing essential citations. For readers interested in these additional aspects, we refer to our recent review (Wagner et al., Nature, 2025), in which the roles of ILC2s and CD8⁺ T cells are discussed in detail.

I congratulate the authors on this refreshing perspective.

Response: Thank you.

Referee #2:

In this commentary, Dr. Marek Wagner and Shigeo Koyasu described the similarity of helminth infection and cancer development regarding tissue invasion, tissue remodeling, and immune cell suppression that can result from persistent cancer progression and helminth infection. The authors noted that both helminth and cancer cells secrete multiple classes of proteases, including metalloproteinases, to degrade critical components of extracellular matrices (ECMs) for their tissue evasion and migration; both promoted angiogenesis, and used blood flow as oxygen and nutrients intake that usually caused bleeding and iron-deficiency anemia; both can develop a type 2 immune response microenvironment including elevations in IL-33, TSLP, IL-4/IL-13, and eosinophilia. Eosinophils can play an important role against helminth infection and cancer development through secretion of toxic enzymes such as major basic protein (MBP).

*The authors also summarized the mechanisms that helminth and cancer cells used to evade immune surveillance including activation and expansion of Treg cells; upregulation of immune checkpoint molecules, such as PD-1 and CTLA-4; as well as secreted immunomodulatory proteins such as TGF-beta mimic from *H. polygyrus*; and active shedding of surface peptides by helminth and tumors. Finally, the authors summarized the study of anthelmintic drugs for cancer therapy potential based on the recognition of functional and metabolic parallels between cancer cells and parasitic helminths. For example, they discuss the use of mebendazole, a benzimidazole derivatives, for therapies of multiple cancers.*

Although this is an excellent start, the authors need to broaden their comparisons of immune cells in helminth infection and cancer. In particular, myeloid cells including macrophages and neutrophils, need to be discussed. Papers, including recent

reviews, by Loke, Maizels (review JEM 2023), Allen, Gause, and Pearce need to be included to provide a more in-depth and more balanced discussion. Of note, a paper published in 2021 in Trends In Immunology (10.1016/j.it.2020.11.006) also discussed similarities between the immune response to helminth infection and cancer. This manuscript, among others, needs to be discussed in the context of this current perspective.

As such this commentary, although of considerable interest, needs significant revisions before it is ready for publication.

Response: We thank the reviewer for the constructive feedback. We fully agree that myeloid cells, particularly macrophages and neutrophils, play important roles in both helminth infection and cancer. However, since this article was submitted as a Perspective rather than a Systematic review, and because the journal enforces a strict 50-reference limit, which we have already exceeded at 69, we are unable to expand the manuscript without removing core citations that support the central argument. Therefore, we have added a brief clarification noting that myeloid cells constitute an additional shared component of helminth and tumor immunity. We also refer interested readers to the most relevant literature for more information while remaining within the citation limit.

Referee #3:

This review compares similarities in the immune response that can be elicited by both cancer and parasitic worm infections. This is an interesting comparison as the review notes that both cancer cells and worms take advantage of similar mechanisms to facilitate its survival and evade immune detection. Given the rapid proliferative capacity of cancer cells and their acquisition of mutations, it is understandable how there is evolutionary convergence for cancer cells to take advantage of similar weaknesses in the immune system as worms. I have included a few comments to hopefully help improve the review.

- Page 8 end of first paragraph, is something missing? This sentence looks like it was cut short.

Response: Thank you for pointing this out. The sentence has now been completed in the revised manuscript.

- The text associated with Figure 3 seems to be missing. Maybe this got accidentally deleted?

Response: Thank you for noticing this. The missing figure text has now been restored.

- A figure showing how anthelmintic drugs can target both helminths and cancer cell pathways may help illustrate how therapeutic targets identified against worms can also have anti-tumour effects.

Response: We agree with this helpful suggestion and have now added a figure illustrating how anthelmintic drugs affect helminths and their potential relevance to cancer pathways.

Dear Dr. Koyasu,

Congratulations on an excellent manuscript! I am very pleased to inform you that your Perspective has been accepted for publication in The EMBO Journal. Thank you for comprehensively addressing the initially raised referee concerns and our editorial requests for changes and re-formatting.

Your manuscript will be processed for publication by EMBO Press. It will be copy edited and you will receive page proofs prior to publication.

Yours sincerely,

Please note that it is The EMBO Journal policy for the transcript of the editorial process (containing referee reports and your response letters) to be published as an online supplement to each paper. If you should prefer removal of any referee-only figures included in the point-by-point response(s), e.g. because they may still be used for future publication or because they have been reproduced from published work by others, please do let us know immediately via response email.

More information is available here: https://www.embopress.org/transparent-process#Review_Process
